# Differential Effect of Non-Thermal Plasma RONS on Two Human Leukemic Cell Populations

**DOI:** 10.3390/cancers13102437

**Published:** 2021-05-18

**Authors:** Hager Mohamed, Eric Gebski, Rufranshell Reyes, Samuel Beane, Brian Wigdahl, Fred C. Krebs, Katharina Stapelmann, Vandana Miller

**Affiliations:** 1Department of Microbiology and Immunology, and Institute for Molecular Medicine and Infectious Disease, Drexel University College of Medicine, Philadelphia, PA 19102, USA; hm469@drexel.edu (H.M.); bw45@drexel.edu (B.W.); fck23@drexel.edu (F.C.K.); 2Department of Biology, College of Arts and Sciences, Drexel University, Philadelphia, PA 19104, USA; ebg42@drexel.edu (E.G.); rmr343@drexel.edu (R.R.); sob29@drexel.edu (S.B.); 3Department of Nuclear Engineering, North Carolina State University, Raleigh, NC 27695, USA; kstapel@ncsu.edu

**Keywords:** oxidative stress, damage-associated molecular patterns (DAMPs), calreticulin, heat shock proteins, phagocytosis, plasma medicine, immunotherapy, hydrogen peroxide, nitrite

## Abstract

**Simple Summary:**

As the number of investigations into the use of non-thermal plasma (NTP) for cancer treatment expands, it is becoming apparent that susceptibility of different cancer cells to NTP varies. We hypothesized that such differences could be attributed to the cell type-dependent interactions between NTP-generated reactive oxygen and nitrogen species (RONS) and the target cells. To test this hypothesis, we examined how two different human leukemic cell lines—Jurkat T lymphocytes and THP-1 monocytes—influence hydrogen peroxide and nitrite content in media after NTP exposure. We also assessed the potential of NTP to enhance immunogenicity in these cells and assayed phagocytosis of NTP-exposed leukemic cells by macrophages. Our results highlight the significance of target-mediated modulation of plasma chemical species in the development and clinical use of protocols involving plasma sources for use in cancer therapeutic application.

**Abstract:**

Non-thermal plasma application to cancer cells is known to induce oxidative stress, cytotoxicity and indirect immunostimulatory effects on antigen presenting cells (APCs). The purpose of this study was to evaluate the responses of two leukemic cell lines—Jurkat T lymphocytes and THP-1 monocytes—to NTP-generated reactive oxygen and nitrogen species (RONS). Both cell types depleted hydrogen peroxide, but THP-1 cells neutralized it almost immediately. Jurkat cells transiently blunted the frequency-dependent increase in nitrite concentrations in contrast to THP-1 cells, which exhibited no immediate effect. A direct relationship between frequency-dependent cytotoxicity and mitochondrial superoxide was observed only in Jurkat cells. Jurkat cells were very responsive to NTP in their display of calreticulin and heat shock proteins 70 and 90. In contrast, THP-1 cells were minimally responsive or unresponsive. Despite no NTP-dependent decrease in cell surface display of CD47 in either cell line, both cell types induced migration of and phagocytosis by APCs. Our results demonstrate that cells modulate the RONS-mediated changes in liquid chemistry, and, importantly, the resultant immunomodulatory effects of NTP can be independent of NTP-induced cytotoxicity.

## 1. Introduction

Non-thermal plasma (NTP) is under active investigation as a broad-spectrum anti-cancer tool due to its dual effect against cancer cells—direct cytotoxicity and indirect stimulation of anti-cancer immune responses [1]. Its efficacy has been demonstrated in animal models of a variety of cancers, including breast cancer, pancreatic cancer, colorectal cancer and melanoma [2,3,4,5]. Current dogma attributes the anti-cancer effects of NTP to reactive oxygen and nitrogen (RONS) species, which are considered to be the active components of NTP that induce oxidative stress in cells exposed to NTP [6].

Because the mechanisms by which NTP affects cancer cells are incompletely understood, ongoing investigations are focused on the identification and characterization of specific physical and chemical plasma effectors generated by NTP, the direct targets of NTP-induced oxidation and secondary effects that are indirectly triggered by NTP [7,8,9]. While animal models have helped identify the different cell populations that are engaged to produce the anti-cancer effects of NTP, the cause-and-effect relationship between NTP effector(s) and discrete cellular outcomes is yet to be defined. This is partially because the cellular responses occur independently of the plasma source used [10]. Because plasma-conditioned liquids produce similar changes in cancer cells as observed in vitro and in vivo after direct treatment by NTP, some effects attributed to NTP may be exclusively chemically triggered [11,12,13,14]. Although the compositions of NTP produced by different plasma-generating devices are comprised of different mixtures of physical and chemical effectors, the operating parameters of most devices can be optimized to achieve similar biological outcomes in target cells, suggesting common mechanisms of action [7]. Finally, the contribution of physical factors during direct treatment with NTP cannot be excluded.

The effects of NTP on cancer cells are also likely influenced by the aqueous environment that surrounds the cancer cells. Tumors surgically exposed in preparation for the application of NTP are not dry, but rather coated in a thin layer of extracellular liquid. Relevant to the focus of this investigation on hematological (blood) cancers, NTP-based treatments of neoplastic cells of immune system origin would likely be conducted with cells suspended in blood plasma or a suitable tissue culture medium. Changes in the chemistry of NTP due to the presence of liquid may alter the effects of NTP on cancer cells as well as surrounding non-cancerous cells.

NTP produces a plethora of RONS in the gas phase that can enter the liquid through the plasma–liquid interface. The plasma-produced RONS undergo further reactions in the gas phase or after entering the liquid phase. The full set of species and reactions is complex and can only be fully addressed in computational models. Lietz and Kushner performed chemistry analysis for a DBD operated in humid air with a liquid (water) layer on top of biological tissue [15]. In their model, 79 gas phase species and 83 liquid phase species were included with 1680 gas phase reactions and 448 liquid phase reactions. Although this model is quite complex, it helps to understand, in a simplified way, the chemistry that the cells would experience when they are exposed to NTP in culture medium. Other components in cell culture medium (e.g., amino acids) provide reaction partners for the reactive species entering the liquid, further altering the chemistry. For example, the presence of the amino acid cysteine has been shown to alter the NO_2_^−^/NO_3_^−^ ratio with considerably higher concentrations of NO_2_^−^ in the presence of cysteine [16,17]. Furthermore, the presence of cysteine reduced the concentration of hydrogen peroxide in the liquid [17]. Yan et al. investigated the stability of NTP effectors in Dulbecco’s modified eagle’s medium (DMEM) after plasma exposure or in solutions prepared using specific components of DMEM. It was observed that the hydrogen peroxide concentration decreased over time, with the largest decrease in cysteine and methionine solutions [18]. In their review, Solè-Martí et al. provided an overview of the interactions of plasma RONS with different liquids being used in plasma cancer research [19]. These studies emphasize the importance of considering the liquid environment in understanding the effects of NTP on mammalian cells.

In addition to the soluble components, the cells themselves provide molecular targets for further reactions. The impact of NTP on the cell is influenced by the composition and the amount of RONS produced, as well as the cell morphology. Various simulations [20,21] indicate that reactive species, particularly OH, react with cell-associated biomolecules, which in turn reduce the OH concentration available for further reactions, including the formation of hydrogen peroxide. These processes can create reactive sites on the biomolecules themselves, corresponding to a transfer of reactivity from the plasma-produced species to the biological molecules. Biomolecules with broken bonds or radical sites could then initiate a cascade of other reactions. Furthermore, cleavage of bonds can lead to the detachment of molecules, providing additional soluble substrates for reactions in the liquid [21]. Collectively, these studies strongly suggest that the target itself can modify at least the chemical composition of plasma through uptake or other chemical reactions, depending on the nature of the target. This is an important point to consider when the target is biological, as is the case in various uses of NTP in the field of plasma medicine. This is a particularly important consideration in the development of NTP-based cancer therapies.

The identity of the cell must also be considered in understanding the effects of NTP on cancer cells. Cells and tissues differ in their capacities to react to oxidants and oxidative stress, dependent on their roles in the body. Some cells are better equipped to counter the effects of oxidative stress caused by intracellular and extracellular RONS. Therefore, as we deliberate the use of NTP for treatment of cancers of different organ systems, the ability of a particular cell type to counter the effectors of NTP cannot be overlooked.

Our current studies were conducted to address alterations in NTP liquid chemistry and NTP-induced biological effects attributable to the presence of cells during NTP application and the identity of the cell exposed to NTP. For these investigations, two human immortalized cell lines were selected: THP-1 monocytes and Jurkat T lymphocytes. As leukemic cell lines with different cell type origins, these cell lines were well suited for comparative studies of the impact of cell phenotype on the effects of NTP applied to blood-derived cancer cells. THP-1 cells are of myeloid origin, have the capacity to produce cellular RONS that are used for destroying pathogens and also have cellular self-protective mechanisms that shield them against the damaging effects of the RONS [22,23]. Macrophages, while not fully resistant to ROS-induced death, survive in oxidative stress environments for long durations via multiple protective mechanisms, including bioactive lipid mediators, nuclear erythroid-derived factor 2 (Nrf2) signaling and metabolic reprogramming. There is also some suggestion that bone marrow-derived M1 macrophages are more resistant to ROS than M2 macrophages [24]. Sensing of ROS by the cytosolic kinases Mst1 and Mst2 may also activate macrophages, leading to inhibited ubiquitination and degradation of Nrf2 that protects cells against oxidative damage [22]. Jurkat cells are T lymphocytes that have a lower capacity to handle increased ROS accumulation [25]. We measured the changes in hydrogen peroxide and nitrite concentrations in the medium in the absence or presence of each cell line over time, evaluated the direct effects of NTP on the cells as well as their abilities to engage antigen presenting cells (APCs).

Our experiments showed that leukemic monocytes quickly depleted hydrogen peroxide from the medium and remained largely resistant to direct cytotoxic effects of NTP. Leukemic T lymphocytes were less efficient at countering hydrogen peroxide and responded more robustly to NTP application. Despite these differences, however, both cell types were similarly subject to phagocytosis by APCs. These observations underscore the importance of NTP liquid chemistry, cellular responses to NTP exposure and downstream immunological mechanisms in developing a complete understanding of the effects of NTP on cancer cells, particularly those involved in hematological malignancies.

## 2. Materials and Methods

### 2.1. Cell Culture

Jurkat cells (clone E6) (ATCC^®^TIB-152™) and THP-1 cells (ATCC^®^TIB-202™) were cultured in RPMI-1640 containing L-Glutamine, 10% fetal bovine serum (FBS) and 1% penicillin/streptomycin. Cells were maintained in an incubator at 37 °C with 5% CO_2_ and 95% humidity.

### 2.2. Non-Thermal Plasma Application

A nanosecond-pulsed dielectric barrier discharge (nsDBD) was used for experiments with an applied voltage of 29 kV, a pulse width of 20 ns and a fixed treatment time of 10 s. The plasma dose was varied by changing the pulse frequency (30, 45, 60, 75, 90 or 105 Hz), based on our previous publications [2,8]. Data points at 0 Hz represent results obtained from cells that were not exposed to NTP.

For experiments involving NTP application to cells, Jurkat or THP-1 cell suspensions (50 µL at 4 × 10^6^ cells/mL) were aliquoted into each well of a 24-well plate to seed 200,000 cells per well. Cells were exposed to the DBD plasma using a 24-well electrode positioned 2 mm above the bottom of the well (Figure 1). After NTP exposure, each well was immediately supplemented with 450 µL of RPMI to establish a final cell density of 4 × 10^5^ cells/mL. For treatment with the antioxidant, N-acetylcysteine (NAC), cells were incubated with 5 mM NAC, (the highest concentration at which cell viability remained unaltered) for 1 h prior to NTP exposure. Cells were then washed once with PBS, re-suspended in RPMI prior to NTP application and then supplemented with 450 µL of RPMI containing 5 mM NAC immediately following NTP exposure.

### 2.3. Hydrogen Peroxide (H_2_O_2_) Detection Assay

H_2_O_2_ was detected using a colorimetric assay utilizing a phenanthroline derivative copper ion method (Catalog No. 11879, Millipore Sigma, Burlington, MA, USA). Immediately after NTP exposure of all samples (approximately 30 min) or after 24 h incubation, 264 µL of NTP-exposed serum-free and phenol red-free RPMI or supernatant from centrifuged suspensions of Jurkat cells or THP-1 cells were added to 36 µL of hydrogen peroxide-based reagent in a 96-well plate as described by the manufacturer (Millipore Sigma). Following a 15 min incubation at room temperature, absorbance of each sample was read at 450 nm using a spectrophotometer (Multiskan Ascent, Thermo Lab Systems; Software Version 2.6). To determine the concentration of H_2_O_2_ in NTP-exposed RPMI, a standard curve was generated using a series of dilutions (from 141.12 to 1.10 µM) in RPMI starting with a 30% weight/weight H_2_O_2_ solution (Catalog No. 1065490100, Millipore Sigma, Burlington, MA, USA) in RPMI.

### 2.4. Nitrite (NO_2_) Detection Assay

Detection of NO_2_ was done using a colorimetric Griess Reagent assay procedure as described by the manufacturer (Catalog No. 109713, Millipore Sigma, Burlington, MA, USA). Immediately after NTP exposure of all samples (approximately 30 min) or after 24 h incubation, 150 µL of NTP-exposed serum-free and phenol red-free RPMI or supernatants from centrifuged Jurkat or THP-1 suspensions were added to wells of a 96-well plate containing 20 µL of Griess Reagent and 130 µL of DI water. Following a 30 min incubation at room temperature, absorbance of the sample was read at 540 nm using a spectrophotometer (Multiskan Ascent, Thermo Lab Systems; Software Version 2.6). To determine the concentration of NO_2_ in plasma-conditioned media, a standard curve was generated using discrete concentrations (100 µM to 0.53 µM) of sodium nitrite (NaNO_2_) (Catalog No. HX0635-3, Millipore Sigma, Burlington, MA, USA) achieved by dilution in RPMI.

### 2.5. Hydroxyl (−OH) Radical Detection Assay

The hydroxyl (−OH) radical detection assay is dependent on hydroxylation of terephthalic acid (TA, ACROS Organics™) to produce the fluorescent 2-hydroxyterepthalic acid (2-HTA, TCI America™). Fifty microliters of a 50 mM TA and 100 mM NaOH solution in serum-free and phenol red-free RPMI were exposed to NTP and immediately supplemented with 450 µL of serum-free and phenol red-free RPMI. 2-HTA quantification was performed using a fluorometric plate reader (Fluoroskan Ascent, Thermo Lab Systems; Software Version 2.6) at an excitation wavelength of 355 nm and an emission wavelength of 460 nm. To determine concentration of -OH in plasma-treated media, a standard curve was generated using a series of dilutions (200 µM to 3.13 µM).

### 2.6. Viability Assay

Jurkat or THP-1 cells exposed to NTP were incubated for 24 h, collected and washed twice with PBS, and stained with 250 nM DAPI (BioLegend, San Diego, CA, USA) for 5 min in the dark before viability was measured using a BD LSRFortessa™ flow cytometer. The percent viable (DAPI negative) cells for each sample was determined using FlowJo™ Software (Beckton, Dickson and Company, Ashland, OR, USA) after doublet exclusion using forward scatter (FSC-A vs. FSC-H).

### 2.7. Surface Marker Analysis

Jurkat or THP-1 cells exposed to either NTP or 250 nM mitoxantrone (MTX) were incubated for 24 h at 37 °C with 5% CO_2_ and 95% humidity. Cells were washed twice with PBS and incubated in blocking buffer containing TruStain FcX (BioLegend, San Diego, CA, USA) for 20 min. Cells were then were stained with antibodies specific to calreticulin (AF750, Novus Biologicals, Littleton, CO, USA), HSP70 (3A3) (AF594, Novus Biologicals, Littleton, CO, USA), HSP90 (AF647, ThermoFisher Scientific, Waltham, MA, USA) and CD47 (AF700, BioLegend, San Diego, CA, USA) on ice for 20 min in the dark. After washing cells twice with PBS, surface marker expression was measured using a BD LSRFortessa™ flow cytometer. Percent positive populations and mean fluorescence intensities (MFI) were calculated using FlowJo™ Software after doublet exclusion using forward scatter (FSC-A vs. FSC-H).

### 2.8. Phagocytosis Assay

THP-1 monocytes were differentiated into mature macrophages (M0) by incubation with 100 nM PMA for 4 days before co-culture studies. Jurkat or THP-1 cells were exposed to NTP, incubated for 24 h at 37 °C with 5% CO_2_ and 95% humidity, washed and then labeled with Hoechst nuclear dye (Catalog No. 62249, ThermoFisher Scientific, Waltham, MA, USA). M0 macrophages were labeled with wheat germ agglutinin (WGA) (Catalog No. W32466, Invitrogen, Carlsbad, CA, USA) and co-cultured with NTP-exposed cells (n = 3 for each condition) at a 1:1 E:T (effector: target) ratio for 1 h at 37 °C. M0 macrophages were washed three times to remove non-internalized NTP-exposed cells at the end of this incubation period. Co-culture of NTP-exposed cells with M0 macrophages at 4 °C for 1 h served as a negative control for phagocytosis. The percentage of M0 macrophages that had engulfed labeled Jurkat or THP-1 cells was determined using a BD LSRFortessa™ flow cytometer. Data were analyzed using FlowJo™ Software after doublet exclusion using forward scatter (FSC-A vs. FSC-H).

### 2.9. Migration Assay

Jurkat or THP-1 cells were seeded into 24-well plates and exposed to NTP. Immediately following exposure, transwell inserts (3 µm pore size) were positioned in each well (n = 3 for each condition) and loaded with THP-1 monocytes pre-labeled with 0.5 µM cell trace far red (CTFR). The number of CTFR-positive THP-1 cells that migrated into the basolateral medium in each well was measured via flow cytometry 24 h later using a BD LSRFortessa™ flow cytometer. Data were analyzed using FlowJo™ Software after doublet exclusion using forward scatter (FSC-A vs. FSC-H).

### 2.10. Statistical Analysis

Data are representative of at least three independent experiments (n ≥ 3 each) unless otherwise indicated. Graphing and statistical analysis were performed using Prism 9 (GraphPad Software, La Jolla, CA, USA). Mean and standard errors were calculated using unpaired Student’s *t*-tests for experiments measuring RONS concentrations and using two-way ANOVA for comparative analysis between samples containing RPMI in the absence of cells and samples containing Jurkat or THP-1 cells. Mean and standard errors were calculated using the Brown–Forsythe one-way ANOVA and Dunnett’s post-hoc test for viability and MitoSOX experiments. Mean and standard errors for surface marker display experiments were calculated using the Kruskal–Wallis test with Dunnett’s post-hoc test.

## 3. Results

### 3.1. The Presence of Cells Alters Plasma-Generated Hydrogen Peroxide and Nitrite Concentrations in Medium

Most studies focus their investigations on the changes produced in cells in response to NTP. However, there is limited understanding of how the presence of cells in media changes the concentration of reactive oxygen species (ROS) and reactive nitrogen species (RNS) in RPMI, at least as measured in the form of peroxide and nitrite, respectively. To examine the effect of the presence of cells on NTP-generated RONS, Jurkat or THP-1 cells were suspended in RPMI and exposed to NTP. To establish comparative data, NTP was applied using the same parameters to RPMI without cells. Colorimetric assays were performed to detect H_2_O_2_ and nitrite in the medium either immediately after treatment (T0) or 24 h after NTP exposure (T24). These time points were selected to evaluate the changes in media peroxide and nitrite levels over time and to establish correlations with biological changes observed in the cells.

When RPMI alone was exposed to plasma, statistically significant increases in both nitrite and hydrogen peroxide were observed (Figure 2). Hydrogen peroxide concentrations (Figure 2a) in NTP-exposed RPMI at T0 and T24 ranged from 35.08 µM at 30 Hz to 74.27 µM at 105 Hz at T0, rising to 50.49 and 59.44 µM at T24, respectively. At T0, the nitrite concentrations (Figure 2b) in NTP-exposed RPMI alone increased from 5.95 µM after 30 Hz exposure to 23.63 µM after 105 Hz exposure. By T24, the concentrations increased to 12.12 and 29.55 µM, respectively. The increases in peroxide and nitrite concentrations in RPMI alone were thus primarily frequency-dependent and persistent during the 24 h post-exposure incubation at 37 °C. In the presence of Jurkat cells, the measured hydrogen peroxide content in RPMI at T0 was much lower than in RPMI alone (ranging from 19.08 µM at 30 Hz to 43.24 µM at 105 Hz) and became depleted to negligible levels by T24 (ranging from 8.34 to 9.93 µM). In contrast, the measured nitrite content at T0 in the presence of Jurkat cells was reduced only at high frequencies (90 and 105 Hz) relative to the nitrite content in the absence of cells (14.26 µM in RPMI with Jurkat cells compared to 19.53 µM in RPMI alone at 90 Hz and 13.90 µM in RPMI with Jurkat cells compared to 23.63 µM in RPMI at 105 Hz). There was no diminution of nitrite concentrations by Jurkat cells at T24; reductions in nitrite concentrations at high frequencies at T0 were not evident at T24 (ranging from 8.05 µM at 30 Hz to 26.13 µM at 105 Hz).

Changes in NTP-dependent nitrite and hydrogen peroxide concentrations in the presence of THP-1 cells differed from those observed with Jurkat cells. At T0, increases in hydrogen peroxide concentrations were negligible or considerably diminished in the presence of THP-1 cells and were only modestly frequency-dependent (5.80 µM at 30 Hz to 10.88 µM at 105 Hz), with significant increases noted only at frequencies of 60 Hz (8.64 µM) and higher (Figure 2a). Similarly, hydrogen peroxide concentrations at T24 were considerably lower in the presence of THP-1 cells (13.23 µM compared to 59.44 µM in RPMI only at 105 Hz) and not frequency-dependent. Nitrite concentrations in the presence of THP-1 cells (Figure 2b) were frequency-dependent at T0 (increasing from 6.09 µM at 30 Hz to 20.00 µM at 105 Hz) and T24 (increasing from 7.70 µM at 30 Hz to 18.78 µM at 105 Hz), similar between T0 and T24 and comparable to concentrations detected in media in the absence of THP-1 cells at T0 (5.95 µM at 30 Hz to 23.63 µM at 105 Hz in RPMI without cells). By T24, nitrite concentrations in the presence of THP-1 cells were still frequency-dependent but somewhat lower than concentrations measured in the absence of cells (18.78 µM compared to 29.55 µM at 105 Hz in RPMI without cells). These results show that NTP-associated peroxide and nitrite chemistries in media are cell type-dependent.

### 3.2. Lymphocytic and Myeloid Leukemic Cells Differ in Susceptibility to NTP

Differences in peroxide and nitrite chemistry in media attributable to the presence of either cell type suggested that cellular responses to NTP exposure would also differ between Jurkat and THP-1 cells. To address this hypothesis, we measured the NTP-dependent cytotoxicity in both cell lines. Previous studies demonstrated that application of NTP to Jurkat cells resulted in increased oxidative stress and stimulation of pro-apoptotic pathways [26,27,28]. In agreement with these studies, we observed that exposure of Jurkat cells to nsDBD plasma induced a frequency-dependent decrease in cell viability, with considerable cytotoxicity observed at 105 Hz (Figure 3a). Consistent with the assumption that NTP-induced cytotoxicity is due to oxidative stress, pre-treatment with the antioxidant N-acetyl cysteine (NAC) ameliorated this cytotoxicity.

In stark contrast, THP-1 cell viability was unaffected by NTP exposure over the range of frequencies used in these experiments. Additional experiments designed to identify frequencies that were capable of inducing cytotoxicity in THP-1 cells demonstrated that THP-1 cell viability was unaffected even after exposure to frequencies as high as 600 Hz (data not shown). This differential response to NTP exposure between Jurkat and THP-1 cells has been reported previously [29]. THP-1 tolerance to NTP did not appear to reflect a generally higher tolerance of those cells to adverse agents, since reductions in THP-1 and Jurkat viabilities were comparable in the presence of mitoxantrone (MTX), which is a clinically approved chemotherapeutic drug. THP-1 tolerance to NTP exposure was paralleled by considerable reductions in peroxides in the presence of THP-1 cells (Figure 2), suggesting a role for mechanisms that protect THP-1 cells from oxidative stress. However, such speculation will need to be substantiated by more detailed analyses involving the use of inhibitors and scavengers to establish specific effectors and mechanisms of action that underlie the negligible impact of NTP on THP-1 viability.

In addition to changes in viability, many cell types respond to plasma RONS by increasing mitochondrial superoxide production as a precursor to oxidative endoplasmic reticulum (ER) stress, as has been reported in several studies using a variety of cell lines [30,31,32,33,34,35]. Therefore, we next assayed for mitochondrial superoxide production in both cell types as another measure of susceptibility to NTP-mediated oxidative stress. Mitochondrial superoxide production was assayed through intracellular staining with MitoSOX red dye and evaluated by epifluorescence using flow cytometry. Consistent with the frequency-dependent decrease in viability, NTP frequency-dependent increases were observed in the percentage of MitoSOX-positive Jurkat cells and levels of cellular MitoSOX fluorescence intensity (mean fluorescence intensity or MFI) in comparison to cells not exposed to NTP (Figure 3b). MitoSOX levels were partially reduced by the presence of NAC, indicating that NTP-induced oxidative stress plays a role in superoxide production.

In these experiments, superoxide production in THP-1 cells in response to NTP exposure was very modest (Figure 3b). Increases in MitoSOX-positive cells within the 30–105 Hz frequency range were less than three-fold. In contrast, NTP application to Jurkat cells resulted in a 10-fold increase in MitoSOX-positive cells at 105 Hz. Similarly, increases in MitoSOX MFI in THP-1 cells exposed to NTP were significant but still less than two-fold over control cells not exposed to NTP. Interestingly, introduction of NAC appeared to have no effect on superoxide production. These data are also consistent with the hypothesis that THP-1 cells have more effective mechanisms for managing oxidative stress in comparison to Jurkat cells.

### 3.3. NTP Stimulates the Display of Pro-Phagocytic Markers on Jurkat Cells at a Higher Magnitude Than on THP-1 Cells

The observed increases in oxidative stress in cell lines in response to NTP exposure served as an impetus for us to investigate downstream consequences of cellular stress. Our next experiments focused on the induction of immunogenic cell death (ICD), which has been demonstrated in cancer cells as an outcome of NTP exposure [2,5,36].

NTP has been shown to enhance immunogenicity of cancer cells by stimulating the emission of various damage-associated molecular patterns (DAMPs) known to promote the function of antigen presenting cells (APC). Of these DAMPs, calreticulin (CRT) is key for promoting phagocytic uptake of cancer cells by APCs. We investigated the extracellular membrane display of CRT, as well as the display of two other pro-phagocytic markers commonly reported after treatment with chemotherapeutic agents: heat shock proteins (HSP) 70 and 90 [37,38,39].

As in previous experiments, considerable differences were observed between Jurkat and THP-1 cell responses to NTP exposure (Figure 4). Application of NTP to Jurkat cells resulted in statistically significant and frequency-dependent increases in the display of CRT (Figure 4a), HSP70 (Figure 4b) and HSP90 (Figure 4c) 24 h after NTP exposure. Increases in DAMP-positive Jurkat cells were considerable, with increases in CRT, HSP70 and HSP90 after exposure to NTP at 105 Hz ranging from approximately 15- to 30-fold above levels observed in cells not exposed to NTP. Notably, the level of CRT displayed on Jurkat cells (Figure 4a) was also greatly increased by exposure to NTP (>100-fold increase in MFI at 105 Hz). Increases in DAMPs appeared to be largely driven by oxidative stress, as suggested by large reductions in CRT, HSP70 and HSP90 by the presence of NAC during NTP exposure. As a control, MTX, which is a potent inducer of ICD [40], also induced significant and large increases in CRT, HSP70 and HSP90 display on Jurkat cells (Appendix A).

In contrast, NTP application to THP-1 cells had little effect on the display of the selected DAMPs. No statistical increases were noted in either CRT (Figure 4a) or HSP90 (Figure 4c). While small increases in HSP70 display were found to be significant and frequency-dependent, the increases were less than three-fold over control levels. Limited responses to NTP are not indicative of an inability of THP-1 cells to respond to external stress-inducing agents, as displayed levels of all three DAMPs were induced significantly and robustly subsequent to MTX exposure (Appendix A).

### 3.4. Emission of DAMPs on NTP-Exposed Jurkat and THP-1 Cells Stimulates Macrophage Function as Measured by Phagocytosis and Cell Migration

A key function of DAMPs is to recruit and engage APCs. Cancer cells, on the other hand, upregulate the expression of the “don’t-eat-me” molecule CD47 on their surface to avoid phagocytic uptake by APCs despite high surface CRT [41,42]. Recently, it was reported that NTP reduces cell surface CD47 by oxidation of the molecule immediately after exposure [43]. To determine if a similar decrease results from NTP application to leukemic cells, the cell surface display of CD47 on Jurkat and THP-1 cells exposed to NTP was examined. In contrast to the previous report, no significant decrease in the display of CD47 on Jurkat or THP-1 cells was observed at T0 relative to the untreated control (Figure 5a). Furthermore, no significant changes in CD47 expression were noted at T24 (Figure 5b).

The absence of any effect of NTP on CD47, which is an opposing signal for phagocytosis [44], will need to be investigated further to reconcile our results with previously demonstrated NTP-induced reductions of CD47 and to determine potential roles for CD47 in immunological therapies based on NTP. However, increases in pro-phagocytic markers on Jurkat cells in response to NTP did imply that NTP exposure would increase the frequency of phagocytosis of Jurkat cells by APCs. Limited induction of DAMPs on THP-1 cells further suggested that phagocytosis of THP-1 cells would be minimally promoted by NTP exposure. To test these hypotheses, NTP-exposed Jurkat or THP-1 cells were co-cultured with M0 macrophages (derived from THP-1 cells). Phagocytosis was measured as the percentage of macrophages that engulfed the NTP-exposed cells at the end of 1 h incubation. Despite the presence of the “don’t-eat-me” molecule CD47, phagocytosis of Jurkat cells by M0 macrophages was robustly stimulated by NTP exposure, with an over a four-fold increase in phagocytosis of Jurkat cells exposed to NTP (105 Hz) relative to the phagocytosis of Jurkat cells not exposed to NTP (Figure 6). Consistent with the little or no induction of DAMP display on THP-1 cells after NTP exposure, a significant but small increase in THP-1 phagocytosis was observed after exposure to NTP at 105 Hz. However, NTP at 30 Hz did not increase phagocytosis relative to THP-1 cells not exposed to NTP.

Cells undergoing ICD also secrete or release DAMPs that serve as chemotactic molecules, resulting in recruitment of APCs. The release of these DAMPs, which include HMGB1 and ATP, after NTP exposure is well documented [2,8,45]. NTP-exposed cells have also been shown to release other pro-chemotactic cytokines, such as IFN-γ and TNF-α, that contribute to increased monocyte migration [45]. To assess the functional effect of DAMPs released from NTP-exposed cells the migration of monocytes across a semipermeable transwell membrane was measured in response to co-culture with NTP-exposed cells. In these experiments, THP-1 cells in the absence of NTP were seeded into the apical chamber as responding cells. NTP-exposed Jurkat or THP-1 cells were seeded immediately after NTP exposure into the basolateral chamber and were used as a potential source of released DAMPs that would provide the chemotactic stimulus. The number of monocytes that migrated into the basolateral chamber was measured after 24 h of co-culture.

We observed that monocyte migration was significantly stimulated in response to both NTP-exposed Jurkat cells and THP-1 cells (Figure 7). Comparing the levels of monocyte migration in response to cells exposed to NTP at 105 Hz, NTP-exposed Jurkat cells more effectively stimulated cell migration (~3-fold increase) relative to NTP-exposed THP-1 cells (<2-fold increase). Interestingly, MTX application to the source cells in the basolateral chamber had a negligible effect on cell migration despite the robust induction of DAMPs on both cell types subsequent to MTX exposure (Appendix A).

## 4. Discussion

These comparative studies were built on the use of two cell lines that have been widely used in investigations that ranged from the development of anti-leukemia therapeutics to modeling immune cell functions in other disease systems. Leukemias are cancers of leucocytes or white blood cells and may be of myeloid (precursors of macrophages and dendritic cells) or lymphoid (precursors of T and B lymphocytes) origin. Cells of the Jurkat cell line are human leukemic cells that originated as CD4-positive T lymphocytes, while THP-1 cells are human myeloid leukemia cells that were derived from a patient with acute monocytic leukemia and are often used for studying innate immune functions. Both cell types are cultured as suspension cells. However, THP-1 cells can be differentiated to adherent, phagocytosis-capable macrophages (M0) and further polarized by appropriate cytokines to function as pro- or anti-inflammatory cells (M1 or M2, respectively) [46].

Previous studies involving these cell lines have reported differences in their responses to NTP. Investigations involving Jurkat cells demonstrated NTP-mediated cytotoxicity and increased cellular stress in these lymphoblastic leukemia cells [28,47]. NTP exposure was shown to stimulate the intrinsic apoptotic pathway in these cells through activation of caspase-8 and upregulation of pro-apoptotic proteins [26,27,28,47]. Genotoxic effects of NTP were also demonstrated through inhibition of cell cycle progression and increased micronuclei in NTP-exposed Jurkat cells, which were reversed when plasma RONS were scavenged by NAC [26]. On the other hand, THP-1 cells have been shown to be quite resistant to the direct cytotoxic effects of NTP [48]. Some reports also demonstrated that NTP induces limited polarization of THP-1 cells in culture [49]. However, most studies primarily focused on modulating the function of THP-1 cells to promote their anti-tumor responses, rather than investigating the therapeutic potential of NTP against leukemic monocytes [29,50,51,52].

Reported differences in the susceptibility of Jurkat and THP-1 cells to NTP have been proposed to be a consequence of multiple factors, including cell type-specific differences in the regulation of pro-apoptotic proteins, baseline antioxidant mechanisms, metabolic activity and even aquaporins and molecules associated with membrane fluidity which can influence RONS uptake [29,48,50,52]. The apparent greater resistance of THP-1 cells to NTP-mediated cytotoxicity may be due to their increased antioxidant capacity, as demonstrated through increased HMOX1, catalase, glutathione oxidases and other enzymes responsible for prevention of oxidative stress [29,52]. This assumption is supported by the observation that cells of monocytic lineage have a robust antioxidant system [22] that provides increased tolerance to the oxidative burst that occurs during phagocytosis. The apparent lack of evidence for a similar level of antioxidant activity associated with Jurkat cells is consistent with greater levels of NTP-associated cytotoxicity and superoxide production in these cells.

In comparing the responses of these two cell lines to NTP, our study indicates that any biological effects measured on NTP-exposed cells are integrations of plasma composition and cellular physiology. For example, THP-1 cells were relatively unresponsive to NTP as measured by cell viability and superoxide production (Figure 2, Figure 3 and Figure 4), presumably due to enhanced antioxidant defense mechanisms available to cells of monocytic lineage. Despite the lack of these NTP effects on THP-1 cells, however, NTP exposure of THP-1 cells still resulted in increased levels of chemotaxis and phagocytosis (Figure 6 and Figure 7), albeit at levels relatively low compared to NTP-exposed Jurkat cells. These results highlight the importance of multi-factor investigations that not only encompass unidirectional interactions between NTP and the target cells, but also a variety of possible NTP-associated outcomes (including cytokine and chemokine release), the complexities of plasma chemistry, and engagement of NTP-exposed cells with other cell types that may secondarily mediate the effects of NTP exposure.

Cell type-dependent differences in responses to NTP highlighted in the present studies may also shed light on the selectivity of proposed NTP-based anti-cancer therapies. Several studies have postulated the existence of an NTP therapeutic window, which is defined by observations that tumor cells (including those that are resistant to the effects of radiation or chemotherapeutic agents) are more susceptible to the cytotoxic effects of NTP relative to non-cancerous cells, which may be more “resistant” to NTP in regimes that have anti-cancer effectiveness [53,54]. The selectivity of NTP has recently been suggested to be due to the long-lived chemical species produced by plasma [55]. Other studies show sparing of the killing of normal cells by NTP, even in the presence of short-lived species, as was the case when cells were directly exposed to dielectric barrier discharge (DBD) plasmas [56]. However, these studies did not consider the ability of specific cells themselves to modify the chemical composition of plasma-exposed medium, which was the focus of our investigations. Mechanisms that underlie the considerable divergence between Jurkat and THP-1 cells in their responses to NTP may also play roles in establishing the apparent selectivity of NTP for cancer cells.

Our results also reveal cell type-dependent differences in NTP-associated RONS chemistry in the media. Changes in the concentrations of RONS over time in the media of NTP-exposed cells may be influenced by a variety of factors, including stability of the species, their release from cells during stress, conversion of these species to other RONS and the formation of these species by other RONS. Hydrogen peroxide may be created by NTP through different pathways. In a humid air discharge, water molecules dissociated in the plasma discharge can recombine to form hydrogen peroxide in the gas phase [15] following the reaction:$${OH}_{g}+{OH}_{g}\;\to H_{2}O_{2g}$$

The gaseous hydrogen peroxide can then enter the liquid where it remains relatively stable. Another pathway relies on precursor species created in the gas phase, such as OH, HO_2_ and H. These can enter the liquid and form hydrogen peroxide in the liquid phase via these reactions:$$H_{aq}+HO_{2aq}\;\to H_{2}O_{2aq}$$
$${OH}_{aq}+{OH}_{aq}\;\to H_{2}O_{2aq}$$

Differences in hydrogen peroxide levels when cells are present during NTP exposure are likely due to the high number of targets on cells for reactions with reactive species, particularly OH, which is a precursor for hydrogen peroxide production. In support of this hypothesis, considerable concentrations of OH were generated in a frequency-dependent manner in RPMI immediately after exposure to NTP (Appendix A). Considering the demonstrated production of both OH and hydrogen peroxide, it is reasonable to assume that OH is quickly consumed in reactions with the cells before it can recombine to form hydrogen peroxide.

The cell type-dependent differences in hydrogen peroxide concentrations in media immediately after NTP exposure in the presence of Jurkat or THP-1 cells are more difficult to reconcile. The morphologies of both cell types are similar, so that a similar OH consumption would be expected for both cell lines. While both cell lines are leukemic, they differ in their origins; Jurkat cells are of lymphoid origin and THP-1 cells are of myeloid origin. Because myeloid cells use cellular RONS to destroy pathogens, they are also equipped with self-protective mechanisms that shield them from the detrimental effects of these reactive species [22,23]. It is therefore tempting to speculate that these self-protective mechanisms are responsible for the lower hydrogen peroxide concentrations in THP-1 cell culture medium compared to concentrations in the presence of Jurkat cells. Plasma ROS that could potentially form H_2_O_2_ or react with biomolecules on the cell surface may be taken up and neutralized by THP-1 cells.

The dominant source of aqueous RNS, on the other hand, is N_x_O_y_ species generated in the gas phase, then solvating into the liquid. As a result, a variety of reactions occur in the liquid, producing acids HNO_xaq_ that subsequently dissociate by hydrolysis. The most prominent reaction products are NO_3_^−^_aq_, NO_2_^−^_aq_ and ONOOH_aq_ [15]. Cluster species, such as HO_2_NO_4aq_, solvate from the gas phase into the liquid and decay to HO_2aq_ and NO_2aq_, with a time constant of several seconds [15]. Other precursors for nitrite include NO_aq_, produced in the gas phase and solvated into the liquid, or ONOOH. A comprehensive reaction list and discussion of the possible reaction pathways can be found in the work of Lietz and Kushner [15].

Nitrite measurements in RPMI in the absence or presence of cells after plasma exposure show higher concentration for all conditions 24 h after treatment compared to directly after treatment. The reaction pathways to create nitrite are rather complex (as discussed above) and some reaction pathways with intermediate transient species will have nitrite or nitrate as more stable end products. The generally lower trends after 24 h for RPMI containing cells compared to the baseline for RPMI alone indicate that the species required to produce nitrite or intermediate reactions are consumed in reactions with the cells or consumed by the cells, similar to mechanisms described above for H_2_O_2_. At higher frequencies, the increase in nitrite is significantly higher for the Jurkat cell culture but remains unchanged for the THP-1 cell culture. This suggests that the THP-1 cells are capable of depleting nitrites by consuming them or potentially neutralizing the species required for the production of nitrite or intermediates that will eventually lead to nitrite production, e.g., OH, NO or ONOOH.

Changes in NTP-associated liquid chemistry in the presence of cells are also indicative of a considerably increased level of complexity in plasma-associated chemistry when cells are considered as active participants in modulating the effects of NTP. Reactions between RONS and cells are added to the array of secondary reactions that NTP-generated RONS can undergo in the gas phase and the media. As shown in molecular dynamics simulations of interactions between peptidoglycans (PG) and plasma-generated ROS [57], ROS in the liquid react with the PG molecules that have not already reacted with RONS. Although PG was chosen to investigate the impact of plasma-treated water on bacteria, the simulations indicate how biomolecules present in the liquid, as well as biomolecules on the cell surface, can deplete and alter the composition of RONS available for further reactions. In a different yet still relevant study, the interactions of ROS with head groups of a phospholipid bilayer were investigated [57]. While HO_2_ and H_2_O_2_ did not lead to bond breaking of the head groups, OH radicals were shown to react with the head groups of the bilayer, leading to cleavage or formation of bonds. While OH produced in the gas or the liquid phase can interact directly with the plasma membrane and specifically with the phospholipid head groups, the oxidation of the head groups is followed by lipid tail oxidation, eventually increasing membrane fluidity. The increased membrane fluidity could then allow RONS to further penetrate through the membrane into the cell interior, where RONS could cause intracellular damage [57].

In addressing the application of NTP to blood cancers, our studies focused not only on cell killing but also on the immunological consequences of NTP exposure. Recently, the emphasis in the field of plasma medicine has been redirected from ablation of tumors by application of NTP toward the application of NTP for cancer immunotherapy [2,5,11]. This approach is based on the observation that NTP helps to overcome the immunological ignorance against the cancer and also directly stimulates innate and adaptive immune responses [2,11,58]. The enhanced immunogenicity by NTP is attributed to induction of ICD as a consequence of plasma-triggered endoplasmic stress. The emission of DAMPs that characterize ICD promotes APC recruitment and function. Many of these DAMPs have been correlated with anti-cancer effects of NTP in vivo. For example, the extracellular translocation of CRT promotes phagocytosis of cancer cells and is reported to be a major contributor of the enhanced protection promoted in vivo by NTP [2]. Our study adds to these observations by showing that NTP-induced stress produces immunomodulatory changes in Jurkat cells, as reflected in the stimulation of monocyte migration and phagocytosis of NTP-exposed Jurkat cells (Figure 6 and Figure 7).

These investigations also demonstrated that NTP-enhanced THP-1 phagocytosis and THP-1-stimulated chemotaxis in the absence of any NTP-associated cytotoxicity. To our knowledge, this is the first report of enhanced immunogenicity of NTP-exposed cells without concomitant cytotoxicity. While this finding has important implications for expanding anti-leukemia immunotherapy options, it also highlights the need to re-evaluate the selectivity of NTP for cancer cells over non-cancerous bystander cells. The absence of NTP-associated cytotoxicity in bystander cells (e.g., overt changes in cell viability or function) during the therapeutic application of NTP may not necessarily signal the absence of unwanted “side effects” due to immunomodulation in those cells.

Our observation that NTP generated at 105 Hz significantly diminished Jurkat cell viability while having no effect on THP-1 viability reflects the challenge in defining the amount or “dose” of NTP necessary to achieve a biological effect, such as tumor ablation or NTP-mediated immunomodulation. NTP dose in plasma medicine encompasses not only the desired biological effects and clinical outcomes of NTP application, but also the avoidance of undesirable side effects. In the delivery of NTP, dose is influenced by NTP composition and the nature of the target that can alter the characteristics of the plasma produced [59]. The major source of controversy is the lack of consensus as to the key effectors—physical, chemical, electrical or some combination. While total plasma chemistry is recognized to correlate reasonably well with biological outcomes for individual devices [7,10,60], cause and effect is difficult to resolve because different plasma devices produce different cocktails of plasma effectors. In addition, and especially in applications involving cancers, biological responses are subject to variabilities in tissue type, moisture content of target, size of the tumor, the tumor microenvironment and other variables, making it difficult to predict therapeutic outcomes and, by extension, the dose of NTP necessary to achieve those outcomes.

## 5. Conclusions

These unique comparative studies focused on the effects of NTP exposure on human leukemic cell lines representative of two different lineages of hematopoietic cells. Our observations indicate that the NTP-mediated increase in immunogenicity of blood cancer cells may be dissociated from NTP-induced cytotoxicity. This observation challenges the simple utility of using cytotoxicity as a first line indicator for NTP-mediated stimulation of desired biological effects across different cell types. Importantly, these studies highlight the need for broader, multi-factorial investigations to develop a more complete understanding of the translational potential of NTP as a therapeutic tool in treating tumors of hematopoietic and lymphoid tissues. Finally, these studies provide a very early foundation for efforts directed toward the development of an ex vivo NTP-based immunotherapy for hematological malignancies.

## Figures and Tables

**Figure 1 cancers-13-02437-f001:**
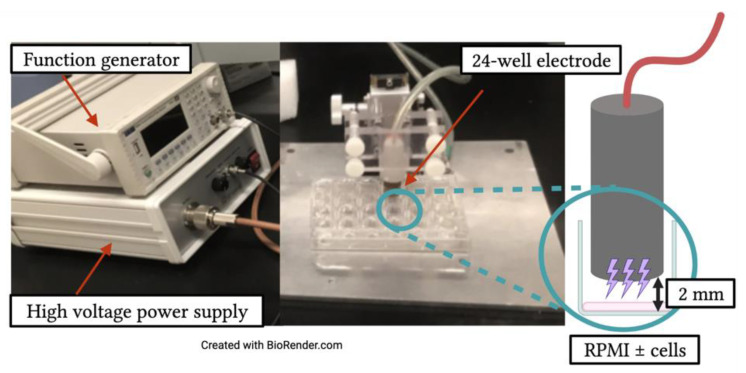
Experimental set-up for exposure of RPMI, Jurkat cells, or THP-1 cells to NTP. A nanosecond-pulsed dielectric barrier discharge (DBD) device was used to deliver NTP to cells or RPMI aliquoted into a 24-well plate. Plasma generation frequency (30–105 Hz) was controlled through the function generator, which was set to deliver a 10 s exposure time. NTP was delivered to cells using a dielectric-encased electrode positioned by a Z-positioner over media in the absence or presence of cells at a distance of 2 mm from the bottom of the well.

**Figure 2 cancers-13-02437-f002:**
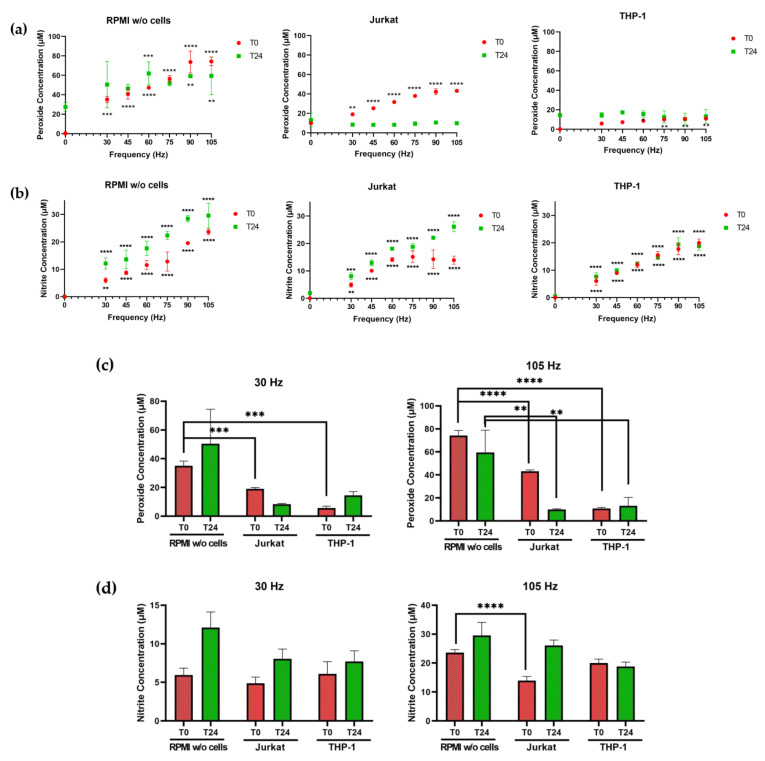
Jurkat or THP-1 cells alter the RONS composition of NTP-exposed media. (**a**) Hydrogen peroxide concentration in RPMI in the absence of cells or containing Jurkat or THP-1 cells and (**b**) nitrite concentration in RPMI without cells or containing Jurkat or THP-1 cells immediately (T0—red) and 24 h (T24—green) after NTP exposure. (**c**) Hydrogen peroxide concentration is lower in the presence of Jurkat or THP-1 cells both at 30 Hz and 105 Hz at T0 and T24 after NTP exposure. (**d**) NTP exposure of RPMI in the absence of cells results in a higher concentration of nitrite in RPMI in the absence of cells than with Jurkat cells, immediately following 105 Hz exposure. Data are presented as mean ± SD from one experiment (n = 3). Significance was calculated using an unpaired Student’s *t*-test (** *p* < 0.01, *** *p* < 0.001, **** *p* < 0.0001).

**Figure 3 cancers-13-02437-f003:**
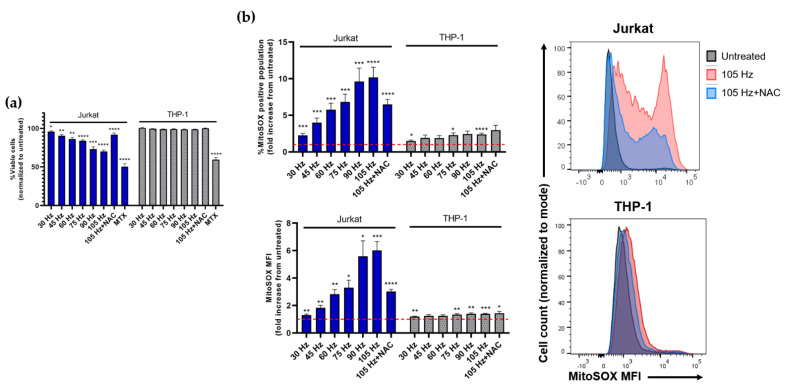
NTP induces cytotoxicity in Jurkat cells but not in THP-1 in a frequency-dependent manner up to 105 Hz. (**a**) Viability of Jurkat and THP-1 cells 24 h post NTP exposure. (**b**) Mitochondrial superoxide production increased in both Jurkat and THP-1 cells 24 h post-NTP exposure, measured both as percent MitoSOX-positive cells and overall mean fluorescence intensity (MFI). Dotted line indicates no change with respect to cells not exposed to NTP. Data are presented as mean ± SEM. Significance was calculated using Brown–Forsythe one-way ANOVA and Dunnett’s post-hoc test (* *p* < 0.05, ** *p* < 0.01, *** *p* < 0.001, **** *p* < 0.0001).

**Figure 4 cancers-13-02437-f004:**
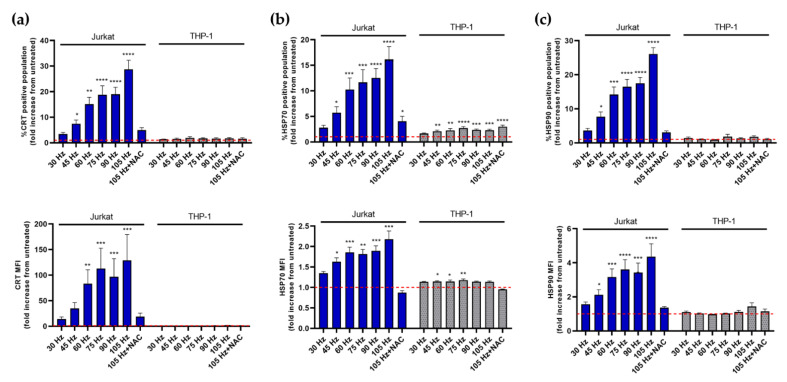
Jurkat and THP-1 cells display pro-phagocytic DAMPs following NTP exposure. The magnitude of increased surface display of (**a**) CRT, (**b**) HSP70, and (**c**) HSP90 differs between NTP-exposed Jurkat and THP-1 cells 24 h post-exposure. Dotted line indicates no change with respect to cells in the absence of NTP. Data are presented as mean ± SEM. Significance was calculated using a Kruskal–Wallis test with Dunnett’s post-hoc test (* *p* < 0.05, ** *p* < 0.01, *** *p* < 0.001, **** *p* < 0.0001).

**Figure 5 cancers-13-02437-f005:**
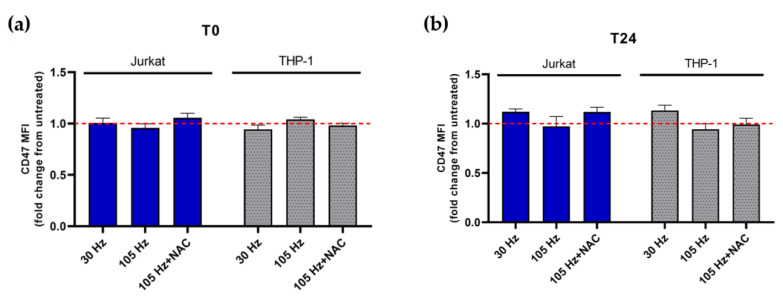
CD47 expression on Jurkat or THP-1 cells is not altered immediately or 24 h post NTP exposure. CD47 expression on Jurkat and THP-1 cells is comparably displayed at both (**a**) 0 h (T0) and (**b**) 24 h (T24) after NTP exposure relative to cells in the absence of NTP. Dotted line indicates no change with respect to cells not exposed to NTP. Data are presented as mean ± SEM. Significance was calculated using a Kruskal–Wallis test with Dunnett’s post-hoc test.

**Figure 6 cancers-13-02437-f006:**
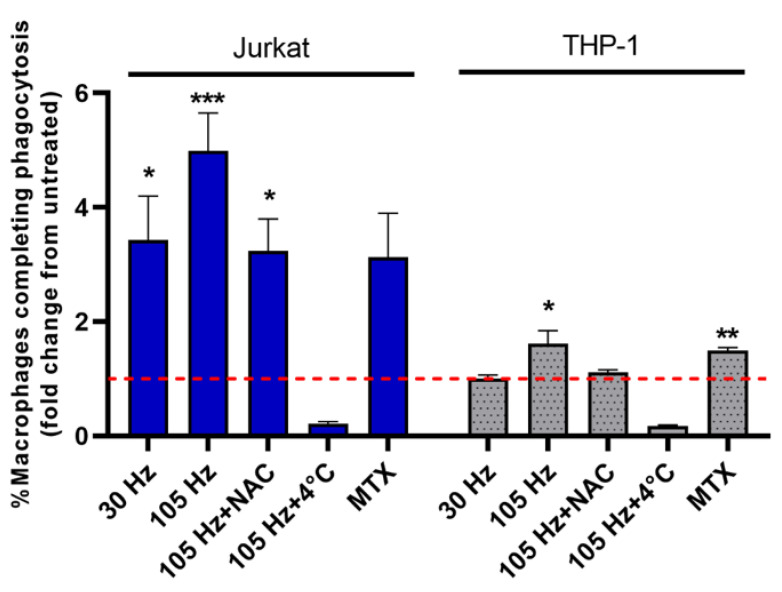
NTP-exposed Jurkat and THP-1 monocytes are phagocytosed by macrophages. Percentage of THP-1 macrophages completing phagocytosis (WGA+HCST+) of Jurkat or THP-1 cells. Dotted line indicates no change with respect to cells in the absence of NTP. Data are presented as mean ± SEM. Significance was calculated using a Kruskal–Wallis test with Dunnett’s post-hoc test (* *p* < 0.05, ** *p* < 0.01, *** *p* < 0.001).

**Figure 7 cancers-13-02437-f007:**
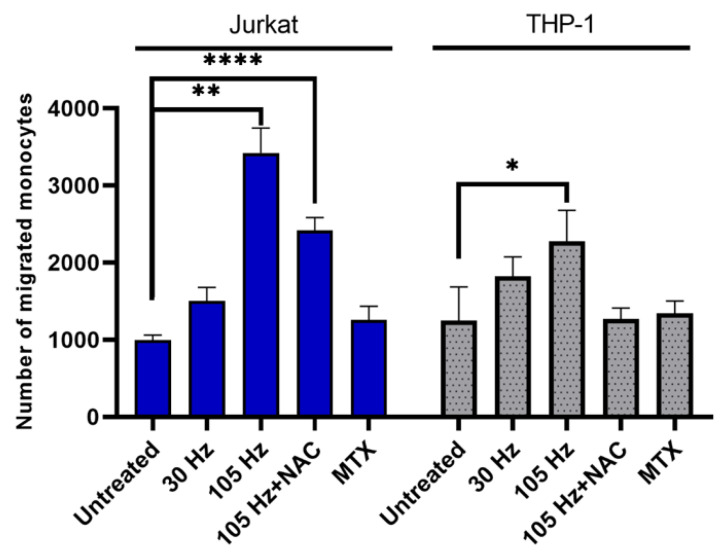
THP-1 monocyte migration is stimulated in response to NTP-exposed cells. The number of migrated (CTFR+) monocytes was increased 24 h post co-culture by NTP-exposed Jurkat or THP-1 cells. Data are presented as mean ± SEM. Significance was calculated using a Kruskal–Wallis test with Dunnett’s post-hoc test (* *p* < 0.05, ** *p* < 0.01, **** *p* < 0.0001).

## Data Availability

The data are contained within the article or the Appendix A.

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
