# Peer review of "Differential Effect of Non-Thermal Plasma RONS on Two Human Leukemic Cell Populations"

_cancers, 2021, doi:10.3390/cancers13102437_

Round 1

Reviewer 1 Report

The authors investigated effects of non-thermal plasma on two different leukemic cell lines: Jurkat cells and THP-1 cells, and demonstrated that immunomodulatory effects of NTP can be independent of NTP-induced cytotoxicity. These achievements are valuable, so I think this manuscript should be accepted with minor revisions.

  1. The authors investigated the frequency dependency, while the other important physical parameters such as exposure time and the distance between the plasma and samples were fixed. I think the authors should describe how they set the parameter, or if the authors investigated the dependency of such other parameters, they should show the data.

  1. The authors insist that THP-1 cells were relatively unresponsive to NTP as measured by cell viability and superoxide production, presumably due to enhanced antioxidant defense mechanisms available to cells of monocytic lineage. I think it is better if the authors can describe more details of the molecular mechanisms that support the idea.

  1. If extracellular hydrogen peroxide and nitrite are main factors, why don’t you add hydrogen peroxide and nitrite instead of NTP-treatment ? This is a typical question in this field, but I think it is better if the authors could discuss.

Author Response

Reviewer 1

The authors investigated effects of non-thermal plasma on two different leukemic cell lines: Jurkat cells and THP-1 cells, and demonstrated that immunomodulatory effects of NTP can be independent of NTP-induced cytotoxicity. These achievements are valuable, so I think this manuscript should be accepted with minor revisions.

  1. The authors investigated the frequency dependency, while the other important physical parameters such as exposure time and the distance between the plasma and samples were fixed. I think the authors should describe how they set the parameter, or if the authors investigated the dependency of such other parameters, they should show the data.

Plasma exposure parameters were selected based on previous publications that describe our work with this nanosecond DBD device (two papers listed below are representative of many relevant publications). The plasma composition can be altered by changing exposure time, frequency, voltage, and/or distance from the sample. In the past, we found that changing the frequency alone (while leaving other parameters constant) was the best way to achieve consistent exposure of all samples, ensuring high reproducibility in cellular responses. Short exposure times, as were used in these studies, preclude a charge buildup that could lead to “streamer memory.”

The revised manuscript includes a statement that the exposure parameters were chosen based on previous studies. This statement also references the papers below.

Lin, A., et al., Nanosecond-Pulsed DBD Plasma-Generated Reactive Oxygen Species Trigger Immunogenic Cell Death in A549 Lung Carcinoma Cells through Intracellular Oxidative Stress. Int J Mol Sci, 2017. 18(5).

Lin, A.G., et al., Non-thermal plasma induces immunogenic cell death in vivo in murine CT26 colorectal tumors. Oncoimmunology, 2018. 7(9): p. e1484978.

  1. The authors insist that THP-1 cells were relatively unresponsive to NTP as measured by cell viability and superoxide production, presumably due to enhanced antioxidant defense mechanisms available to cells of monocytic lineage. I think it is better if the authors can describe more details of the molecular mechanisms that support the idea.

Although a thorough discussion of this topic is beyond the scope of this manuscript., we have added the following statement to expand on our theory about the tolerance of THP-1 cells to NTP exposure:

Macrophages, while not fully resistant to ROS-induced death, survive in oxidative stress environments for long durations via multiple protective mechanisms, including bioactive lipid mediators, nuclear erythroid-derived factor 2 (Nrf2) signaling, and metabolic reprogramming. There is also some suggestion that bone marrow derived M1 macrophages are more resistant to ROS than M2 macrophages (Regdon et al.). Sensing of ROS by the cytosolic kinases Mst1 and Mst2 may also activate macrophages, leading to inhibited ubiquination and degradation of Nrf2 that protects cells against oxidative damage (Wang et al.).

Regdon, Z., et al., LPS protects macrophages from AIF-independent parthanatos by downregulation of PARP1 expression, induction of SOD2 expression, and a metabolic shift to aerobic glycolysis. Free Radic Biol Med, 2019. 131: p. 184-196.

Wang, P., et al., Macrophage achieves self-protection against oxidative stress-induced ageing through the Mst-Nrf2 axis. Nature Communications, 2019. 10(1): p. 755.

  1. If extracellular hydrogen peroxide and nitrite are main factors, why don’t you add hydrogen peroxide and nitrite instead of NTP-treatment? This is a typical question in this field, but I think it is better if the authors could discuss.

Hydrogen peroxide and nitrite are the stable end products of plasma in liquids and are most often reported in literature. While we do not claim that hydrogen peroxide and nitrite are the key effectors, we agree that comparisons with exogenously added hydrogen peroxide and nitrite are very important. This was already investigated as part of another recent publication (Ranieri et al.). Similarly, many other investigators have shown in the past that hydrogen peroxide and nitrite are only partially responsible for the observed cellular effects.

Ranieri, P., et al., GSH Modification as a Marker for Plasma Source and Biological Response Comparison to Plasma Treatment. Applied Sciences, 2020. 10(6))

Reviewer 2 Report

The acute myelogenous leukemia and acute lymphoblastic leukemia are still hard to achieve complete remission. Once relapsed, the treatment for these malignancies is difficult due to the resistance to conventional chemo and radiotherapy, which are used as established first-line therapy. In this manuscript, Mohammad H et al demonstrated non-thermal plasma (NTP) efficiently damaged Jurkat cell lines, which is known as T lymphocyte-derived leukemia. This manuscript is well-organized, however; following points should be clarified.

Major points.

#1. The direct exposure of NTP cause blood coagulation. Authors need to discuss this point for clinical application.

#2. What is the difference between figure 2a,b and 2c,d ? These data sets seems similar result.

Minor points.

#1. In figure 2, authors demonstrates the concentration of peroxide. Are they hydrogen peroxide?

#2. In figure 4a, the Y-axis of calreticulin is expressed as fold change and its value ranges from 0 to 200. Are mean fluorescent intensities elevated one hundred times after 105Hz NTP exposure ?

Author Response

Reviewer 2

The acute myelogenous leukemia and acute lymphoblastic leukemia are still hard to achieve complete remission. Once relapsed, the treatment for these malignancies is difficult due to the resistance to conventional chemo and radiotherapy, which are used as established first-line therapy. In this manuscript, Mohammad H et al demonstrated non-thermal plasma (NTP) efficiently damaged Jurkat cell lines, which is known as T lymphocyte-derived leukemia. This manuscript is well-organized, however; following points should be clarified.

Major points.

  1. The direct exposure of NTP cause blood coagulation. Authors need to discuss this point for clinical application.

The reviewer’s observation is correct with respect to blood coagulation caused by NTP exposure. We would like to clarify that we are not proposing whole blood exposure to NTP as the therapeutic modality for leukemia. While our studies represent very early steps in the development of NTP for immunotherapy of leukemia, we have speculated on the use of NTP for vaccination strategies and as an ex vivo approach for leucocyte disorders, paralleling early efforts directed toward developing a similar ex vivo NTP-based immunotherapy for HIV-1 infection in the following publications (Mohamed et al. 2020, and 2021).

Mohamed, H., et al., Non-thermal plasma modulates cellular markers associated with immunogenicity in a model of latent HIV-1 infection. PLOS ONE, 2021. 16(3): p. e0247125.

Non-thermal plasma as part of a novel strategy for vaccination, Hager Mohamed, Rita A. Esposito, Michele A. Kutzler, Brian Wigdahl, Fred C. Krebs, Vandana Miller, Plasma Processes and Polymers, 2020. 17:e2000051

Nevertheless, at the reviewer’s suggestion, we have added the following statement in our conclusion to alleviate any confusion or concern:

Finally, these studies provide a very early foundation for efforts directed toward the development of an ex vivo NTP-based immunotherapy for hematological malignancies.

  1. What is the difference between figure 2a,b and 2c,d ? These data sets seems similar result.

The data were plotted in two different formats to underscore the temporal differences between the Jurkat and THP-1 cells, specifically at the frequencies where most of the immunological responses were characterized. This allowed us to perform and clearly depict the statistically significant differences in the data between different groups to make it easy for the reader to understand.

Figures 2a and b show the changes in concentrations of the RONS in NTP-exposed RPMI alone or with Jurkat and THP-1 cells over time (T0 and T24) and highlight the statistical significance of exposure to different frequencies as compared to no NTP (0 Hz).

Figures 2c and d depict the correlation between the concentrations in RPMI only and that of RPMI containing Jurkat or THP-1 cells specifically at the frequencies where most of the immunological responses were characterized.

Overall, Figure 2 demonstrates that the cell type present during NTP exposure influences plasma chemistry.

Minor points.

  1. In figure 2, authors demonstrates the concentration of peroxide. Are they hydrogen peroxide?

The reviewer is correct, it is hydrogen peroxide. We have made this correction in the figure legend.

  1. In figure 4a, the Y-axis of calreticulin is expressed as fold change and its value ranges from 0 to 200. Are mean fluorescent intensities elevated one hundred times after 105Hz NTP exposure?

The figure is correct as shown. The graph reflects a large increase in the amount of calreticulin translocated after NTP exposure. Most cells do not display CRT on their surface under normal circumstances, hence the MFI is negligible for non-plasma exposed cells.
